# Stress Prediction Model of Super-High Arch Dams Based on EMD-PSO-GPR Model

**Chunyao Hou [1], Yilun Wei [2], Hongyi Zhang [1], Xuezhou Zhu [2], Dawen Tan [1], Yi Zhou [1] and Yu Hu [2,*]**

[1] Three Gorges Jinsha River Chuanyun Hydropower Development Co., Ltd., Yong Shan Xiluodu Power Plant, Chengdu 657300, China; hou_chunyao@ctg.com.cn (C.H.); zhang_hongyi@ctg.com.cn (H.Z.); tan_dawen@ctg.com.cn (D.T.); zhou_yi7@ctg.com.cn (Y.Z.)

[2] Hydraulic Department, Tsinghua University, Beijing 100084, China; weiyilun@edu.iwhr.com (Y.W.); zhuxz@mails.tsinghua.edu.cn (X.Z.)

[*] Correspondence: huyu93@tsinghua.org.cn

**Abstract:** In response to the challenge of limited model availability for predicting the lifespan of super-high arch dams, a hybrid model named EMD-PSO-GPR (EPR) is proposed in this study. The EPR model leverages Empirical Mode Decomposition (EMD), Gaussian Process Regression (GPR), and Particle Swarm Optimization (PSO) to provide an effective solution for super-high arch dam stress prediction. This research focuses on three strategically selected measurement points within the dam, characterized by complex stress conditions. The predicted results from the EPR are compared with those from GPR, Long Short-Term Memory (LSTM), and Support Vector Regression (SVR), using actual stress data measured at research points within a super-high arch dam in Southwest China. The findings reveal that the proposed EPR model attains a maximum mean absolute error (MAE) of 0.02916 and a maximum root mean square error (RMSE) of 0.03055, surpassing the compared models. As a result, the EPR model introduces an innovative computational framework for stress prediction in super-high arch dams, excelling in handling stress data characterized by high vibration frequencies and providing more accurate predictions.

**Keywords:** super-high arch dam; stress prediction; GPR; EMD; PSO





## 1. Introduction

A critical component of dam safety monitoring is the development of a suitable mathematical model. This model is employed to compare observed data with predicted values, enabling the detection of deviations. This methodology expedites the assessment of possible unusual trends and enables the selection of appropriate corrective actions [1]. Currently, the construction of super-high arch dams is advancing towards increased heights while reducing thickness, underscoring the importance of carefully choosing relevant parameters to establish accurate mathematical models. Numerous engineering practices and associated studies affirm a prevailing consensus that the monitoring of extensive deformation and displacement is inherently more intuitive and effective [2–14]. Deformation, fundamentally, manifests as stress followed by strains across diverse structural sections in response to applied loads. The primary integral of strain corresponds to rotation angles, and the secondary integral yields displacement [15]. Deformation displacement, stemming from multifaceted influences, encompasses various incongruous components. Notably, temperature effects exert substantial influence, with the temperature deformation of concrete arch dams constrained by both structural boundaries and the concrete's properties. Consequently, temperature-induced stress emerges, and unconstrained free temperature deformation remains notably significant [15]. This emphasizes that a significant portion of deformation does not necessarily correlate with elevated stress levels, nor can it be construed as an indicator of dam failure.

Conversely, stress and strain exhibit heightened sensitivity and effectiveness. Structural cracking and deterioration invariably initiate when stress levels surpass prescribed thresholds. Should tensile stress exceed these limits, the dam structure may fracture; similarly, surpassing shear stress limits could lead to dam body slippage. Instances of localized dam failure consistently correspond to anomalous data within stress–strain monitoring datasets [16]. Evidently, stress–strain monitoring assumes paramount importance for concrete arch dams. However, the vast majority of current work by scholars has focused on the study of dam deformation [7,17–19]. Even when focusing on stress analysis, the majority of scholars set the research object as stress analysis of dam bodies under seismic loading [20,21]. There are fewer studies that completely set the research object as the stress analysis of arch dams under normal operation.

Concrete stress–strain monitoring serves the purpose of comprehending the precise stress distribution within the dam and identifying the location, magnitude, and orientation of peak stress. This process aids in gauging the dam's strength and safety, thereby furnishing essential insights for dam operation and reinforcement strategies. Additionally, stress observation outcomes play a role in evaluating the validity of design calculation methodologies, thereby fostering potential enhancements and advancing the state of scientific and technological understanding [22]. Consequently, in practical engineering research, a stress-centric approach becomes imperative, necessitating the development of a fitting stress prediction model grounded in measured data.

Zhou et al. [23] devised an innovative approach by amalgamating monitoring data and finite element analysis to introduce a hybrid prediction model for dam behavior. This model integrates real-time monitoring data with comprehensive simulations of the dam's entire process, facilitating the inclusion of dam deformation and pivotal stress indicators. The resultant early warning index system furnishes distinct thresholds for deformation and stress across various dam segments. Li et al. [8] proposed a numerical analysis method for decoupling the thermal structure of the dam–water–foundation system considering the viscoelastic properties of the material and the internal and external thermal loads, and, from the results, the stress-hazardous area during the first storage period of the extra-high arch dam is mainly concentrated in the heel region of the dam. In another study, Zhang et al. [24] presented a stress–strain model founded upon an Artificial Neural Network (ANN) to predict the behavior of concrete columns under concentric compression, specifically focusing on pressure and related force characteristics. Notably, the field of dam stress prediction remains relatively unexplored. Consequently, this paper also compiles diverse research methodologies for stress prediction across varied structural contexts. Fan et al. [25], for instance, combined chaos theory and backpropagation (BP) neural networks to establish a chaotic time series prediction model. Their model incorporated engineering data and enabled stress prediction and segmental structural analysis by analyzing chaotic data patterns. Similarly, Ma et al. [26] applied finite element numerical simulation to analyze the stress distribution of an underground comprehensive pipe gallery project, establishing a predictive model linking structural stress and ground fissure displacement. Zhang et al. [27], in a different context, introduced a Kriging-based approach to predict the assembly stress of sealing rings. Their method, combining finite element simulations with machine learning, was demonstrated to be suitable for online prediction of assembly stress, revealing correlations between assembly condition parameters and sealing ring stress distribution.

In conclusion, research on stress prediction for super-high arch dams remains limited. Most scholars have relied on conventional statistical models and finite element analyses. However, challenges persist within these traditional approaches: 1. The parameters within traditional statistical and finite element models remain static and lack the ability to dynamically adapt to changing calculation conditions. 2. Both the traditional statistical models and finite element analyses tend to oversimplify the research subject into a "black box" scenario with a fixed internal structure. 3. The presentation format of calculation outcomes lacks the warning capability provided by interval bands, failing to convey the uncertainty associated

with the results. These shortcomings highlight the need for more advanced and adaptable methodologies in stress prediction for super-high arch dams.

To gain a deeper understanding of the operational behavior of super-high arch dams, the development of stress prediction models capable of adapting to external environmental changes is of paramount importance. Consequently, this paper proposes a novel stress prediction model denoted as EMD + PSO + GPR (referred to as EPR model), building upon previous research efforts.

The model primarily takes advantage of the EMD model's ability to handle high-frequency data and the minimum hyperparameters of the GPR model and is optimized by the PSO model to ensure better compatibility with current engineering problems.

In summary, this paper's innovations are highlighted as follows:

1. Introduction of a stress prediction framework, the EPR model, tailored to ultra-high arch dams.
2. Incorporation of deep learning models into the realm of stress prediction for super-high arch dams.

The paper's organizational structure is as follows. The Section 2 outlines the principles and methodologies underlying the model's construction. The Section 3 offers an illustrative example, validating the model's efficacy using a super-high arch dam in Southwest China. Subsequently, the Section 4 presents the model's computational outcomes. Finally, the Section 5 encapsulates the paper's core conclusions.

## 2. Proposed Model

The EPR model has been intricately designed to facilitate stress predictions tailored to super-high arch dams. Section 2.1 meticulously outlines the comprehensive procedure governing the inception and structure of the EPR model. Simultaneously, Section 2.2 offers an insightful explanation of the data preprocessing phase, shedding light on the complex mechanisms involved in converting strain data into accurate stress data. In concise narratives, Sections 2.3 and 2.4 further elaborate on the key components of the EPR model—Empirical Mode Decomposition (EMD) and Gaussian Process Regression (GPR), respectively.

### 2.1. EPR Stress Prediction Model

2.1.1. EPR Stress Prediction Modeling

In this study, we establish a hybrid model by connecting the EMD and GPR models in sequence. It is worth noting that the distribution of the measured stress and strain data series may not strictly adhere to a Gaussian distribution. Consequently, employing the maximum likelihood function for direct parameter estimation may not always yield optimal results. To address this issue, we incorporate an additional PSO model to optimize these parameters. The procedural steps for the developed EPR stress prediction model are depicted in Figure 1.

In this paper, the main stress is predicted, and the results are evaluated using the main tensile stress intensity criterion. The main tensile stress intensity criterion is shown in Equation (1) [28].

$$\sigma_1 \leq [\sigma] \tag{1}$$

where $\sigma_1$ represents the maximum principal stress and $[\sigma]$ represents the permissible stress.

The implementation steps for the proposed stress prediction model for high arch dams outlined in this paper are as follows:

1. **Data collection:** real-time monitoring data from extra-high arch dams are organized based on historical deformation, water level, temperature, and other relevant factors.
2. **Stress–strain data solution:** utilizing the approach detailed in Section 2.1, strain monitoring data is transformed into corresponding stress values.

3. **Data set classification and parameter initialization:** the processed dataset is input into the model, and the data are partitioned into training and validation sets. The training set comprises 75% of the dataset.

4. **Model training and prediction:** the stress solution data are fed into the EMD model for decomposition. High-frequency data are transformed into low-frequency data to ensure a greater number of sub-series adhere to standard function distributions. The low-frequency data are subsequently fed into the GPR model individually, with the PSO model being employed for parameter optimization. The predictions from the sub-series are then combined to produce the final model prediction series.

5. **Model results evaluation:** model performance is assessed using relevant evaluation indices. The model results are output once the values of the model evaluation indices meet predetermined criteria.

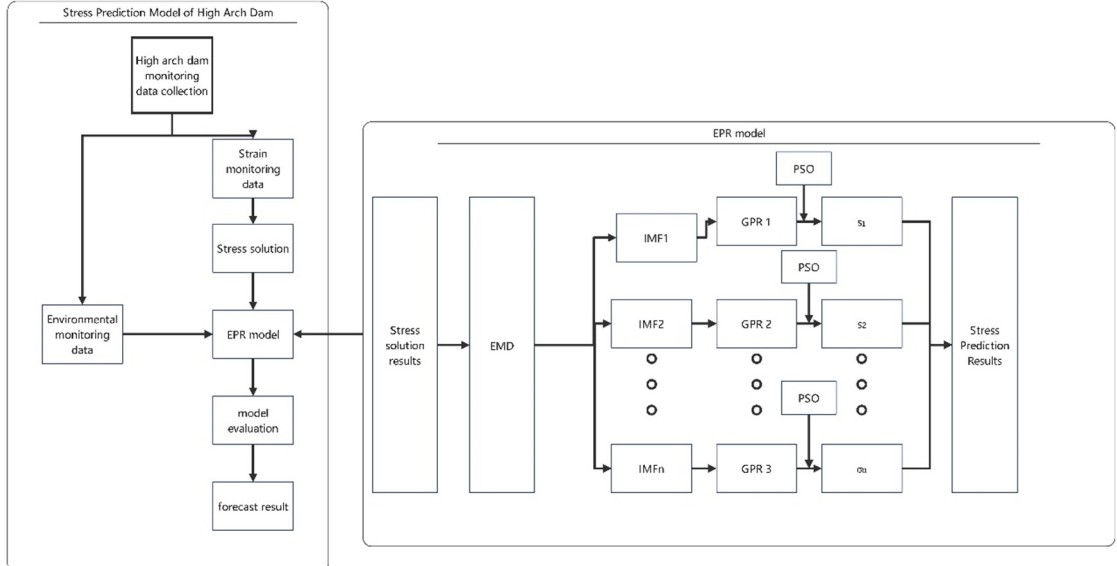

**Figure 1.** EPR stress prediction model diagram.

2.1.2. Model Inputs

According to the theory of Wu Zhongru et al. [29–32], the stresses in concrete dams are mainly related to water pressure, temperature, self-weight, and aging. The mathematical expression is shown in the form of Equation (2) [29–32].

$$\sigma = \sigma_H + \sigma_T + \sigma_G + \sigma_\theta \tag{2}$$

where $\sigma_H$ is the water pressure component; $\sigma_T$ is the temperature component; $\sigma_G$ is the self-weight component; and $\sigma_\theta$ is the aging component. The water pressure component ($\sigma_H$) is expressed as a polynomial in the head component ($H$), as shown in Equation (3) [29].

$$\Sigma_H = \sum_{i=1}^{4} a_i H^i \tag{3}$$

where $H$ indicates the water depth.

1. $\sigma_T$ temperature component:

This component is expressed using the average temperature of the strain gauge set, as shown in Equation (4) [29].

$$\sigma_T = dT \tag{4}$$

where T denotes the average temperature of the strain gauge set.

2. $\sigma_G$ self-weight component:

$\sigma_G$ is constant when the height of the dam is a given. For extra-high arch dams, this is generally calculated using Equation (5) [29].

$$\sigma_G = \sum_{i=1}^{3} b_i W^i \tag{5}$$

$W$ is the thickness of the concrete poured above the measuring point.

3. $\sigma_\theta$ aging component:

The aging component reflects the change in stress due to concrete creep, dry shrinkage, and wet expansion, and is generally calculated by Equation (6) [29].

$$\sigma_\theta = c_0 + c_1 t + c_2 \ln t \tag{6}$$

where $t$ is the age of the concrete, calculated from the starting point of the analysis period.

In summary, the input variables for the model can be categorized into four types of influencing factors as derived from Equations (2)–(6): water level factors (H, $H^2$, $H^3$, $H^4$), temperature factor (T), self-weight factors (W, $W^2$, $W^3$), and time factor (t, ln(t)). As components of the prediction model's input, these eight factors collectively impact dam displacement and can be concisely represented by Equation (7).

$$input = (x_1, x_2, x_3, x_4, x_5, x_6, x_7, x_8) = \left( H, H^2, H^3, H^4, T, W, W^2, W^3, t, lnt \right) \tag{7}$$

*2.2. Stress Computation from Strain Monitoring Data*

This paper aims to utilize the deformation method, a widely adopted approach for addressing strain monitoring results. The deformation method involves dividing the uniaxial strain timeline into discrete periods. Drawing upon the principles of creep behavior, the stress increment from previous periods induces creep deformation within the current period. The cumulative effect of this creep deformation, combined with instantaneous elastic deformation, contributes to the overall deformation [33]. An essential aspect of this approach is considering stress increments from earlier periods when retrospectively calculating along the time axis. The specific calculation process is outlined as follows:

(1) Division of time periods: the uniaxial strain process is divided into n time intervals, which can be either uniformly or non-uniformly distributed. Shorter intervals are preferred during periods with significant early stress increments, while longer intervals are used when stress changes are minimal later on. Starting from a zero-observation point, the first recorded observation marks the first point, along with the corresponding time period. Subsequently, the second observation signifies the second point, and the interval between it and the first point constitutes the second time period. This pattern continues, defining successive time intervals. The beginning and end moments (ages) for each interval are as follows:

$$\tau_0, \tau_1, \tau_2, \ldots, \tau_{i-1}, \tau_i, \ldots, \tau_{n-1}, \tau_n$$

(2) Calculation of total deformation process line: The total deformation process line is established for each time interval based on the loading age. This can be achieved by utilizing data from creep tests or by referring to a table that correlates effective modulus and total deformation with the midpoint age, following the stress increment age. Alternatively, a function that characterizes the degree of creep in relation to loading age and duration can be developed, assisting in further conversion of the effective modulus.

(3) Calculation of incremental stress for the current period: The measured strain at a specific moment encompasses both the elastic deformation resulting from the incremental elastic stress at that point and the cumulative deformation resulting from previous stresses up to that moment. Therefore, when calculating the incremental strain for that specific moment, this cumulative deformation should be subtracted. The cumulative deformation effect preceding the calculated time interval is referred to as "pre-strain" and can be estimated using Equation (8) [33].

$$\varepsilon_h = \sum_{i=0}^{n-1} \Delta\sigma_i \left[ \frac{1}{E(\tau_i)} + c(\overline{\tau}_n, \tau_i) \right] \tag{8}$$

where $\varepsilon_h$ is the forward strain before $\tau_{n-1} \sim \tau_n$ (the current time period) and $\overline{\tau}_n = \frac{\tau_{n-1}+\tau_n}{2}$ is the age at the midpoint of the time period.

The stress increment at the current age should be

$$\Delta\sigma_n = E_s(\overline{\tau}_n, \tau_{n-1}) \left\{ \varepsilon'_n(\overline{\tau}_n) - \sum_{i=0}^{n-1} \Delta\sigma_i \left[ \frac{1}{E(\tau_i)} + c(\overline{\tau}_n, \tau_i) \right] \right\} \tag{9}$$

where $E_s(\overline{\tau}_n, \tau_{n-1})$ is the reciprocal of the total deformation per unit stress continuously applied to $\overline{\tau}_n$ at the age of loading with $\tau_{n-1}$, i.e., the effective modulus of elasticity at the moment of $\overline{\tau}_n$; $\varepsilon_n\prime(\overline{\tau}_n)$ is the uniaxial strain value at the moment of $\overline{\tau}_n$ on the uniaxial strain process line.

(4) Calculation of true stress:

$$\sigma_n = \sum_{i=0}^{n} \Delta\sigma_i \tag{10}$$

### 2.3. Empirical Mode Decomposition

Empirical Mode Decomposition (EMD) is a signal processing technique in the time–frequency domain that relies on the inherent time-scale characteristics of the data, devoid of any predefined basis functions [34]. This method excels even with a high signal-to-noise ratio [35]. Central to this approach is the empirical mode decomposition, which dissects intricate signals into a finite set of intrinsic mode functions (IMFs). Each IMF component encapsulates details about the local attributes of the original signal across various time scales.

EMD decomposes the input signal into multiple intrinsic mode functions and a residual component composed according to Equation (11) [34].

$$I(n) = \sum_{m=1}^{M} IMF_m(n) + Res_M(n)I(n) \tag{11}$$

Here, $I(n)$ represents the input signal, $IMF_m(n)$ represents the $m$th intrinsic mode function, and $Res_M(n)$ represents the residuals.

The procedure for extracting an IMF is referred to as sifting, and the sifting process encompasses the following steps:

(1) Identify the local extreme value points.

(2) Connect the upper envelope using cubic spline interpolation through the extreme value points and likewise for the lower envelope.

(3) Determine the mean value ($m_1$) of the upper and lower envelopes.

(4) Subtract the calculated mean value of the upper and lower envelopes from the input signal.

$$X(t) - m_1 = h_1 X(t) - m_1 = h_1 \tag{12}$$

One iteration of the above process does not guarantee that $h_1$ is an eigenmode function (IMF), and the above process needs to be repeated until $h_1$ is an eigenmode function (IMF).

A single iteration of the aforementioned process does not guarantee that $h_1$ qualifies as an intrinsic mode function (IMF). Hence, the procedure must be reiterated until $h_1$ indeed satisfies the conditions of an eigenmode function (IMF).

The stopping criterion governs the number of sifting iterations undertaken to yield an intrinsic mode function (IMF). Within this paper, the sifting process halts once the standard deviation (SD) [36,37] falls below a predetermined threshold. The computation of SD adheres to the principles delineated in Equation (13) [36,37].

$$SD_k = \sum_{t=0}^{T} \frac{|h_{k-1}(t) - h_k(t)|^2}{h_{t-1}^2(t)} \tag{13}$$

### 2.4. Gaussian Process Regression

Gaussian Process Regression (GPR) is a non-parametric model employed for regression analysis of data, utilizing the Gaussian process (GP) prior [38].

GPR's model assumptions encompass both noise (regression residuals) [38] and Gaussian process [39] priors, which are resolved through Bayesian inference [38]. Without imposing constraints on the kernel function's form, GPR stands theoretically as a versatile approximation for continuous functions within a confined space. Additionally, GPR can offer a posterior for prediction outcomes, and this posterior possesses an analytical structure when the likelihood follows a normal distribution. Thus, GPR emerges as a probabilistic model boasting versatility and resolvability.

Gaussian Process Regression, or GPR, is commonly used in scenarios where there are limited and low sample sizes. However, more advanced algorithms have been developed to handle larger datasets and high-dimensional contexts [40,41]. In the domain of regression prediction, the usual goal is to predict a single-point value. What sets GPR apart is its unique ability to provide probabilistic predictions. This enhances the informative value of predictions by not only providing a precise point prediction but also upper and lower prediction bounds, giving a range of likely outcomes. The Gaussian process regression is outlined in Equation (14) [38].

$$f(x) \sim N\big(\mu(x), k(x, x')\big) \tag{14}$$

$\mu(x)$ denotes the mean function and $k(x, x')$ denotes the covariance function. The covariance function is expressed in the form shown in Equation (15) [38].

$$k(x, x') = \theta_0^2 \exp\left(-\frac{(x - x')^2}{2\theta_1^2}\right) + \sigma^2 \delta_{ij} \tag{15}$$

$\delta_{ij}$ is the Dirac function, with $\delta_{ij} = 1$ when $i = j$ and 0 otherwise.

It is evident that the GPR model features just two hyperparameters, $\theta_0$ and $\theta_1$, and the prediction process can be executed by solving for these values. The detailed operational process of the GPR model is outlined below:

1. Hypothesizing that the observed data, denoted as discrete data $(x_o, y_o)$, and assuming that $y_o$ and $f(x)$ adhere to a joint Gaussian distribution, the expression for the joint probability density is provided by Equation (16) [39].

$$\begin{bmatrix} f(x) \\ y^o \end{bmatrix} \sim N\left(\begin{bmatrix} \mu_f \\ \mu_y \end{bmatrix}, \begin{bmatrix} K_{ff} & K_{fy} \\ K_{fy}^T & K_{yy} \end{bmatrix}\right) K_{ff} = k(x, x), K_{fy} = k(x, x^o), K_{yy} = k(x^o, x^o) \tag{16}$$

$x$ is the predicted independent variable and $x^o$ is the known observed independent variable.

2. Equation (17) can be derived from the Bayesian probability expression.

$$f \sim N\left(K_{fy}^T + \mu_f, K_{yy} - K_{fy}^T K_{ff}^{-1} K_{fy}\right) \tag{17}$$

3. The predicted mean and error matrices are shown in Equations (18) and (19) [39–41].

$$y_{mean} = K_{fy}^T K_{ff}^{-1} y^o \tag{18}$$

$$y_\sigma = K_{yy} - K_{fy}^T K_{ff}^{-1} K_{fy} \tag{19}$$

4. Optimise the hyperparameters in the maximum likelihood expression case. The maximum likelihood expression is shown in Equation (20) [39–41].

$$\text{Log} p(y|\theta_0, \theta_1) = \log N\big(0, K_{yy}(\theta_0, \theta_1)\big) = -\frac{1}{2}y^T K_{yy}^{-1} - \frac{1}{2}\log|K_{yy}| - \frac{1}{2}\log(2\pi) \tag{20}$$

*2.5. Particle Swarm Optimization*

Particle Swarm Optimization (PSO) is a significant population-based intelligence algorithm introduced by Kennedy and Eberhart in 1995. It draws inspiration from simulating bird flock behaviors and has evolved into a crucial branch of evolutionary algorithms. The algorithm commences by initializing a population of random particles, where each particle symbolizes a prospective solution. Over iterations, each particle is guided towards both its own optimal position and the best position within the entire population. Notably, the algorithm is characterized by its intuitive nature, straightforward implementation, and efficient execution [42]. The equations governing the particle's velocity and position updates are detailed in Equations (21) and (22) [42].

$$v_{id}^{k+1} = \omega v_{id}^k + c_1 r_1 \left( p_{id}^k - z_{id}^k \right) + c_2 r_2 \left( p_{gd}^k - z_{id}^k \right) \tag{21}$$

$$z_{id}^{k+1} = z_{id}^k + v_{id}^{k+1} \tag{22}$$

where $z_{id}$ is the $d$ dimensional position vector of the ith particle; $v_{id}$ is the flight velocity of the particle; $p_{id}$ is the optimal position of the particle searched so far; $\omega$ is the inertia weight, indicating the degree of influence of the previous particle's velocity on the current particle; $r_1$, $r_2$ are random numbers between [0, 1]; and $c_1$, $c_2$ are learning factors.

**3. Data Collection and Pre-Analyses**

*3.1. Engineering Background*

This paper validates the proposed methodology using monitoring data collected from the XLD super-high arch dam situated in Southwest China. The XLD dam, a super-high arch dam in the 300 m class category, plays a pivotal role as a critical water storage, flood control, and power generation facility in China. The top arch center line of the XLD super-high arch dam spans a length of 681.51 m. Figure 2 provides a comprehensive overview of the XLD dam.

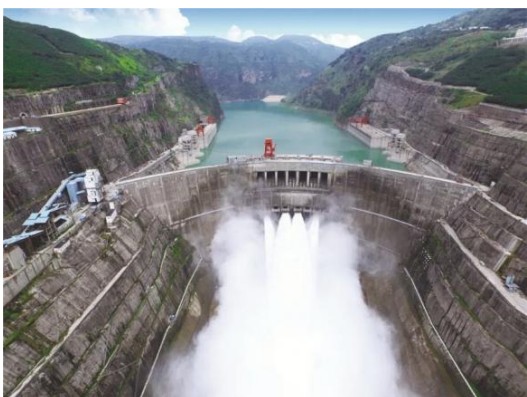

**Figure 2.** Panoramic view of the XLD dam.

*3.2. Data Collection and Pre-Processing*

The XLD super-high arch dam is equipped with a sophisticated intelligent structural safety monitoring system. This system has been meticulously designed to continuously monitor the safety condition of critical components, including the dam body, adjacent bank rock masses, bedrock, and other vital structures. Additionally, it provides real-time monitoring of various factors such as deformation patterns, temperature fluctuations, seepage characteristics, and other relevant loads and their impacts. For the purposes of this study, our focus is primarily on stress data associated with the dam body. As outlined in Section 2.1.2, the inputs for the high arch dam stress prediction model encompass variables such as water level, temperature, self-weight, and aging factors.

Given the inherent material property of concrete, which has a limited tensile capacity, and the heightened risk of crack propagation due to tensile stresses near the dam's heel region [38], our study concentrates on three strategically positioned strain gauges: S67-1, S616-4, and S622-5. These gauges are strategically located in the heel areas of three distinct dam sections, namely 7#, 16#, and 22#. These selections have been made to effectively demonstrate the efficiency of our methodology. Figure 3 provides a clear illustration of the precise locations of these measurement points. Among these sections, 7# and 22# are positioned on the bank slope dam sections, while section 16# is situated on a riverbed dam section and an arch crown beam dam section. Analyzing stress distribution within the dam section's heel across various transverse river directions allows us to explore stress variation mechanisms. This approach contributes significantly to an improved understanding of the comprehensive stress attributes of the high arch dam's heel region.

**upstream**

**downstream**

**Figure 3.** Diagram showing the exact location of the study measurement points.

To portray the spatial stress–strain condition at the designated study site, a six-way strain gauge has been selected for this study. Figure 4 shows a schematic diagram of the 6-way strain gage and a construction drawing of the field burial.

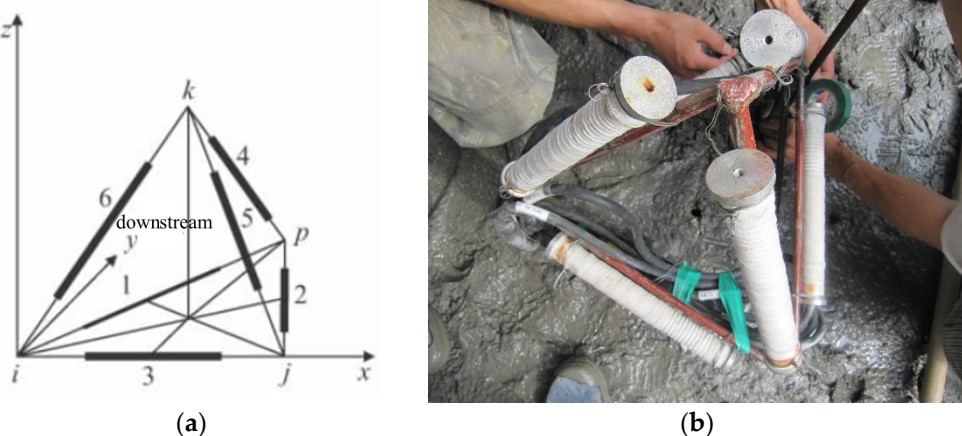

(**a**)                                    (**b**)

**Figure 4.** Schematic diagram of the six-direction strain gauge and on-site construction drawings. (**a**) Diagram of a six-way strain gauge; (**b**) site construction layout plan.

Figure 5 presents a time-series curve that showcases the variation in strain at the designated measurement point, along with the temperature of the dam body at that specific location. Blue-black square lines represent dam temperature, red squares represent

1-direction strain, blue circular lines represent 2-direction strain, light blue triangular lines represent 3-direction strain, pink inverted triangular lines represent 4-direction strain, purple rhombic lines represent 5-direction strain, and green arrow-shaped lines represent 6-direction strain. The graph reveals that during the construction phase, the strain of the dam body is significantly influenced by the temperature of the dam body. Subsequently, during the operational phase, there is a noticeable hysteresis effect in the relationship between the dam body's strain and its temperature. From the figure, it can be seen that after the temperature of the dam concrete stabilizes, the strain of the dam body itself gradually becomes cyclic.

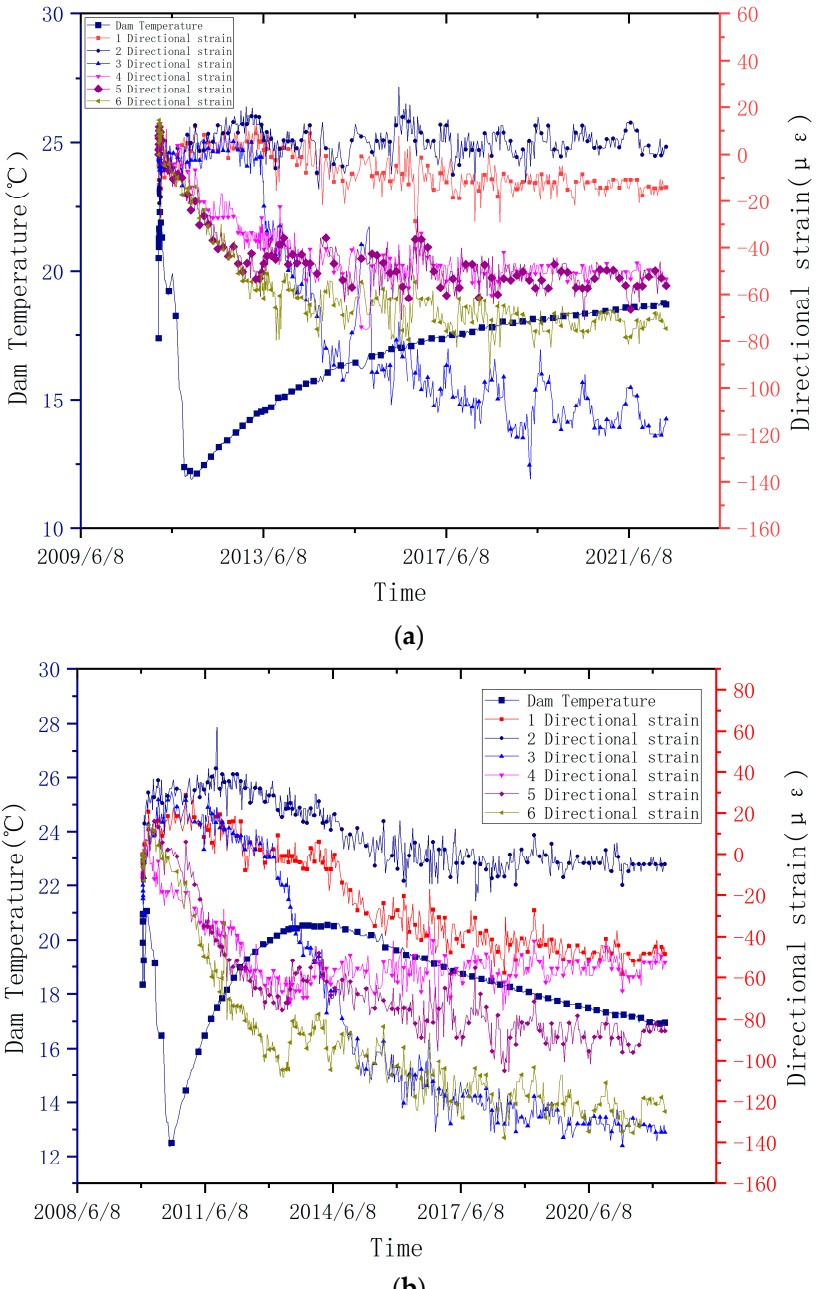

**Figure 5.** *Cont.*

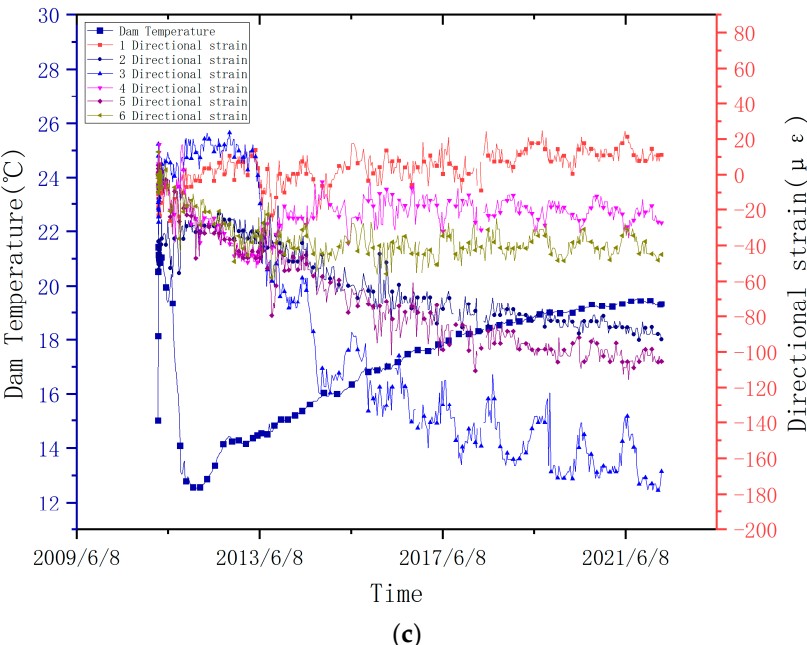

(**c**)

**Figure 5.** Study of strain–temperature time series curves at measurement points. (**a**) S67-1 Measurement point strain–temperature time-series process curve; (**b**) S616-4 measuring point strain–temperature time series process curve; (**c**) S622-5 measuring point strain–temperature time-series process curve.

Figure 6 displays a plot detailing the temporal evolution of strain concerning the upstream water level at the designated measurement points. Blue-black square lines represent upstream water levels, red squares represent 1-direction strain, blue circular lines represent 2-direction strain, light blue triangular lines represent 3-direction strain, pink inverted triangular lines represent 4-direction strain, purple rhombic lines represent 5-direction strain, and green arrow-shaped lines represent 6-direction strain. As the graph illustrates, there exists a positive correlation between the strain experienced by the dam body and the water level upstream. Variations in the water load upstream lead to fluctuations in the strain encountered by the dam body. From the figure, it can be seen that after the dam enters the operation period, the strain of the dam body itself shows a periodic change with the upstream water level.

Figure 7 presents the outcomes of the stress solution observed at the specified measurement points. Conforming to the pertinent principles outlined in Section 2.1, the strains are translated into stresses in six distinct directions. In this particular investigation, the focus is directed towards the positive stress in the $\sigma_{yy}$ direction. From the figure, it can be seen that the stress increases sharply during the construction period, and the stress change tends to stabilize after entering the operation period and shows certain periodic changes.

The XLD dam has undertaken precise real-time monitoring of crucial parameters such as temperature, water level, and strain, accumulating a substantial volume of valuable data. This abundance of data makes it feasible to establish a stress prediction model for high dams. In this study, a total of 3416 data points were selected, covering the period from 9 November 2012 to 22 March 2022 for model construction. Out of these, 2562 data points were allocated for model training, spanning from 9 November 2012 to 8 November 2019. Subsequently, data from 9 November 2019 to 22 March 2022 were set aside for model validation.

Given the diverse data scales present in various types of monitoring data, normalization is applied using Equation (23) [43].

$$Z = \frac{x_i - \mu}{\sigma} \tag{23}$$

where $\mu$ is the mean and $\sigma$ is the standard deviation.

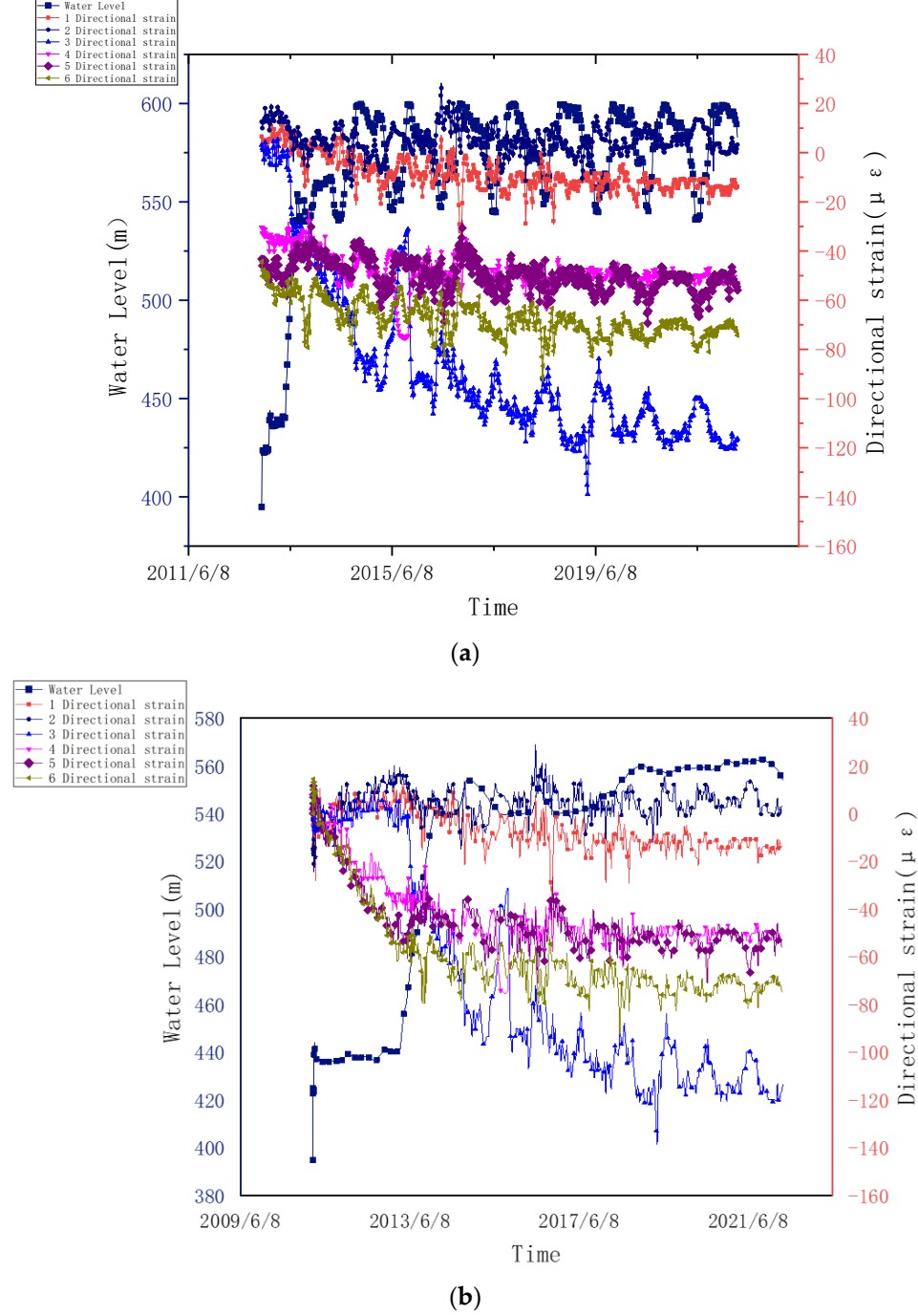

(**a**)

(**b**)

**Figure 6.** *Cont.*

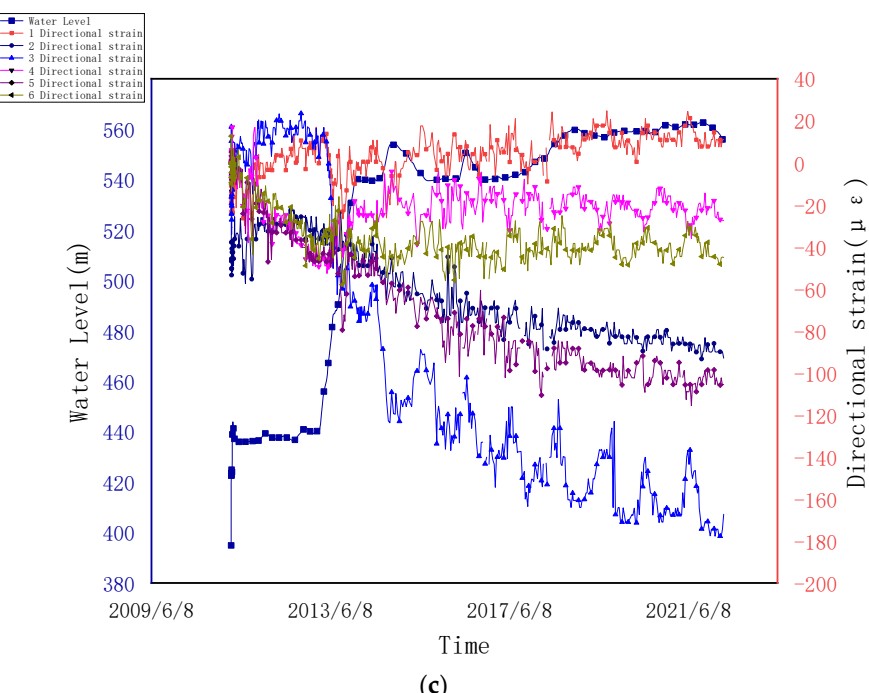

(**c**)

**Figure 6.** Study of strain–upstream water level time-series processes at measurement points. (**a**) S67-1 measurement point strain–upstream water level time course curve; (**b**) S616-4 measuring point strain–upstream water level time course curve; (**c**) S622-5 measuring point strain–upstream water level time course curve.

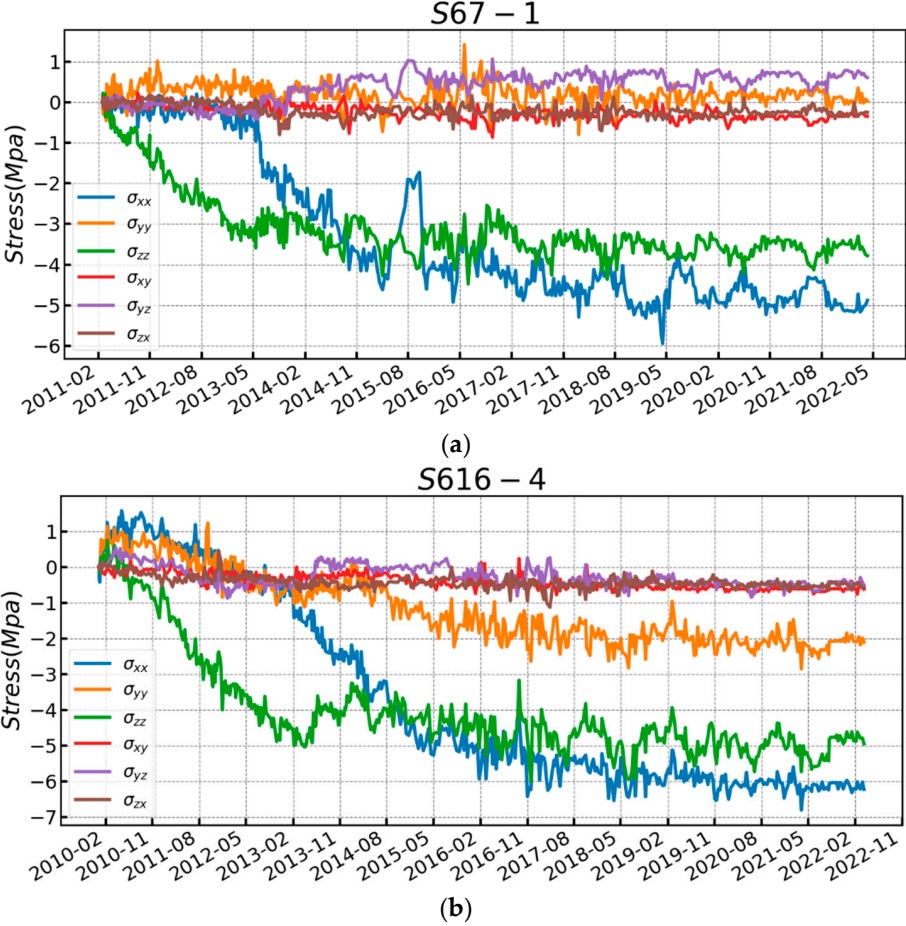

(**a**)

(**b**)

**Figure 7.** *Cont.*

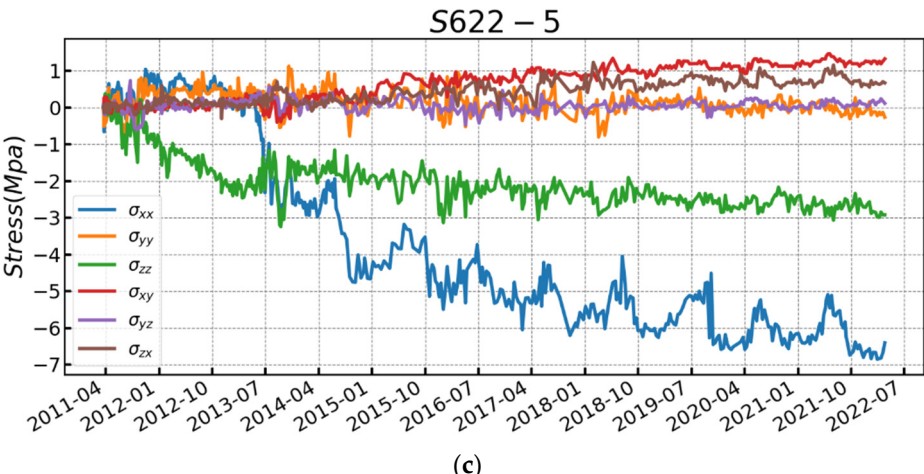

(**c**)

**Figure 7.** Study of stress solution results at measurement points. (**a**) S67-1 stress solution results at measurement point; (**b**) S616-4 stress solution results for measurement points; (**c**) S622-5 stress solution results at measurement points.

### *3.3. Model Evaluation Indicators*

To validate the efficacy of the proposed model, this study employs two widely utilized statistical evaluation metrics: the mean absolute error (MAE) [44–46] and the root mean square error (RMSE) [47–49]. These indicators serve as benchmarks for assessing the model's performance. The precise calculation formulas for MAE and RMSE are provided in Equation (24) [44–46] and Equation (25) [47–49]. Smaller values for both MAE and RMSE indicate closer proximity to 0, signifying enhanced model accuracy.

$$MAE = \frac{1}{n}\sum_{i=1}^{n}|y_i - \hat{y}_i| \tag{24}$$

$$RMSE = \sqrt{\frac{1}{n}\sum_{i=1}^{n}(y_i - \hat{y}_i)^2} \tag{25}$$

where $n$ is the total number of samples to be evaluated and $y_i$ and $\hat{y}_i$ are the measured and predicted displacement values.

## 4. Results and Discussions

In this section, we conduct a comprehensive comparison between the outcomes derived from various contrasting models and our proposed EPR model. Additionally, we present a detailed analysis of the stress data decomposition accomplished by the EMD model.

### *4.1. EMD Model Stress Data Decomposition Results*

Stress data often exhibits characteristics such as nonlinearity, pronounced randomness, and susceptibility to significant noise interference, which can make direct prediction less effective. To address these challenges, this study employs the Empirical Mode Decomposition (EMD) method to decompose stress data into distinct intrinsic mode functions (IMFs) and a residual component, thereby mitigating nonlinear effects.

For the stress data associated with the specified measurement points, this paper utilizes the EMD approach to break them down into multiple IMFs with varying frequencies. In total, seven IMFs are extracted through this process. These decomposed IMFs are visually represented in Figure 8.

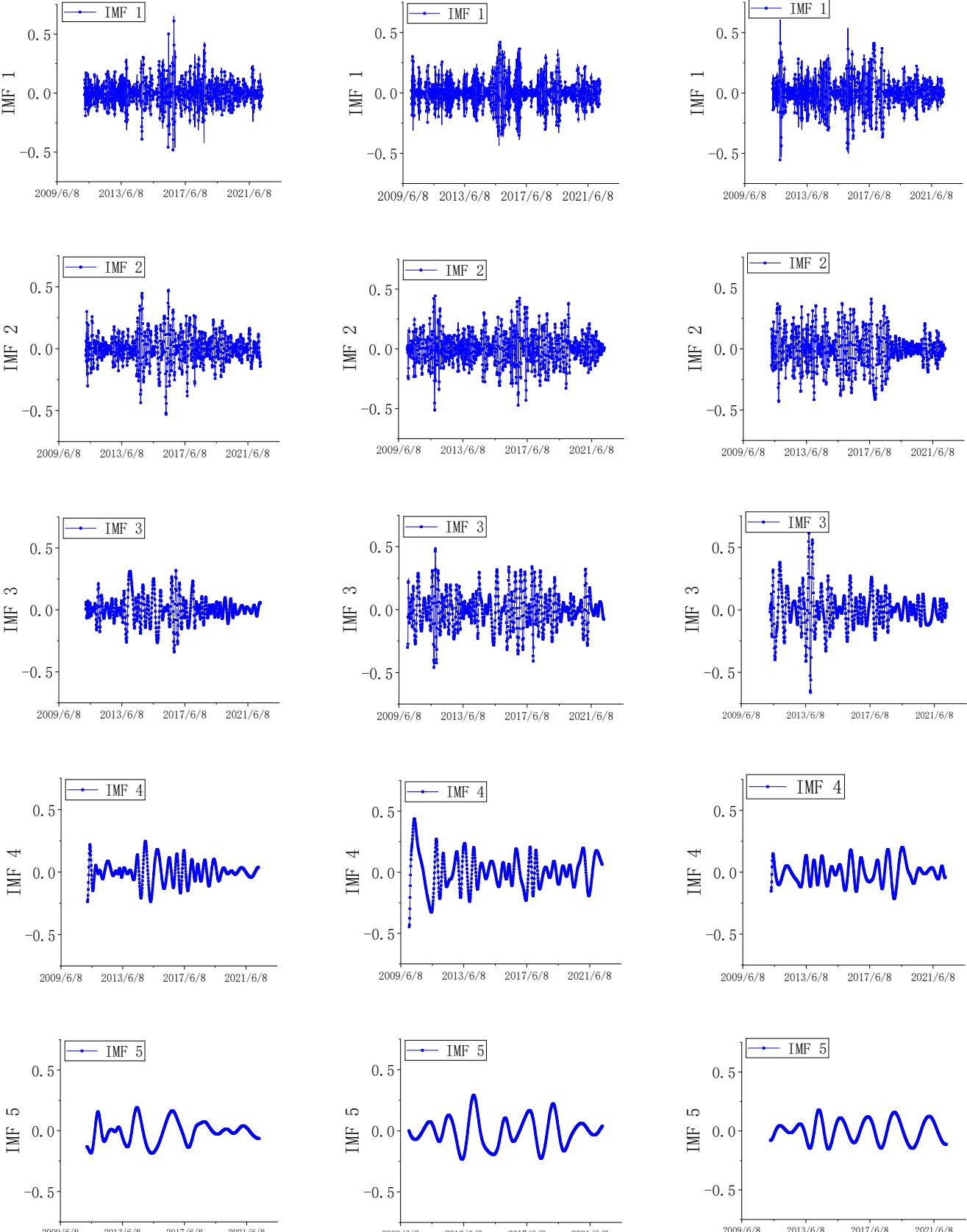

**Figure 8.** *Cont.*

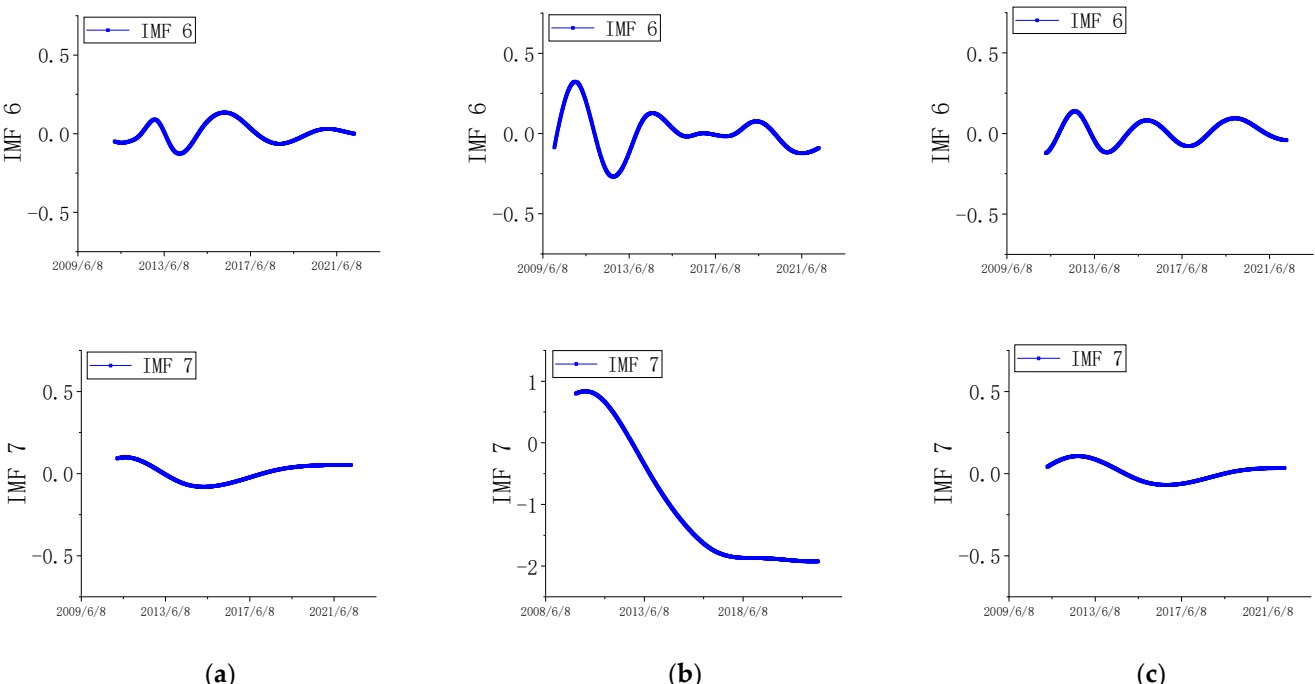

**Figure 8.** EMD decomposition results of research point stress data. (**a**) S67-1 stress decomposition results; (**b**) S616-4 stress decomposition results; (**c**) S622-5 stress decomposition results.

The outcomes of the Empirical Mode Decomposition (EMD) unveil distinct frequency characteristics within the seven decomposed intrinsic mode function (IMF) sequences. If each individual component were directly input into the subsequent computational network, it might not comprehensively capture the data features. To address this, the decomposed IMFs are categorized into two groups: high-frequency and low-frequency data. More precisely, IMF1–IMF4 make up the high-frequency components, while IMF5–IMF7 constitute the low-frequency components. This categorization is intended to enhance feature extraction and improve prediction outcomes.

### 4.2. EPR Model Results

Figure 9a–f provide a side-by-side comparison of the forecasted stress values for the extra-high arch dam as projected by the trained EPR model and the actual measured values. In these figures, the measured values are indicated by the black line, while the model's predicted values are represented in red. The observations from Figure 9 confirm the EPR model's proficiency in predicting stress levels for the dam body of extra-high arch dams. Moreover, the model consistently exhibits strong performance across the prediction of dam sections 7#, 16#, and 22#, highlighting its stability and reliability.

Table 1 offers a comprehensive summary of the results, demonstrating that the EPR model attains a maximum RMSE value of 0.216171 and a maximum MAE value of 0.29167 within the research point prediction dataset. These results highlight the model's capability to provide accurate predictions that align with the acceptable engineering standards. This outcome further solidifies the effectiveness and practicality of the EPR approach for stress prediction in super-high arch dams.

In conclusion, based on the outcomes of our model, it is evident that the EPR model stands as a powerful and competitive tool for forecasting stress levels in the dam bodies of super-high arch dams.

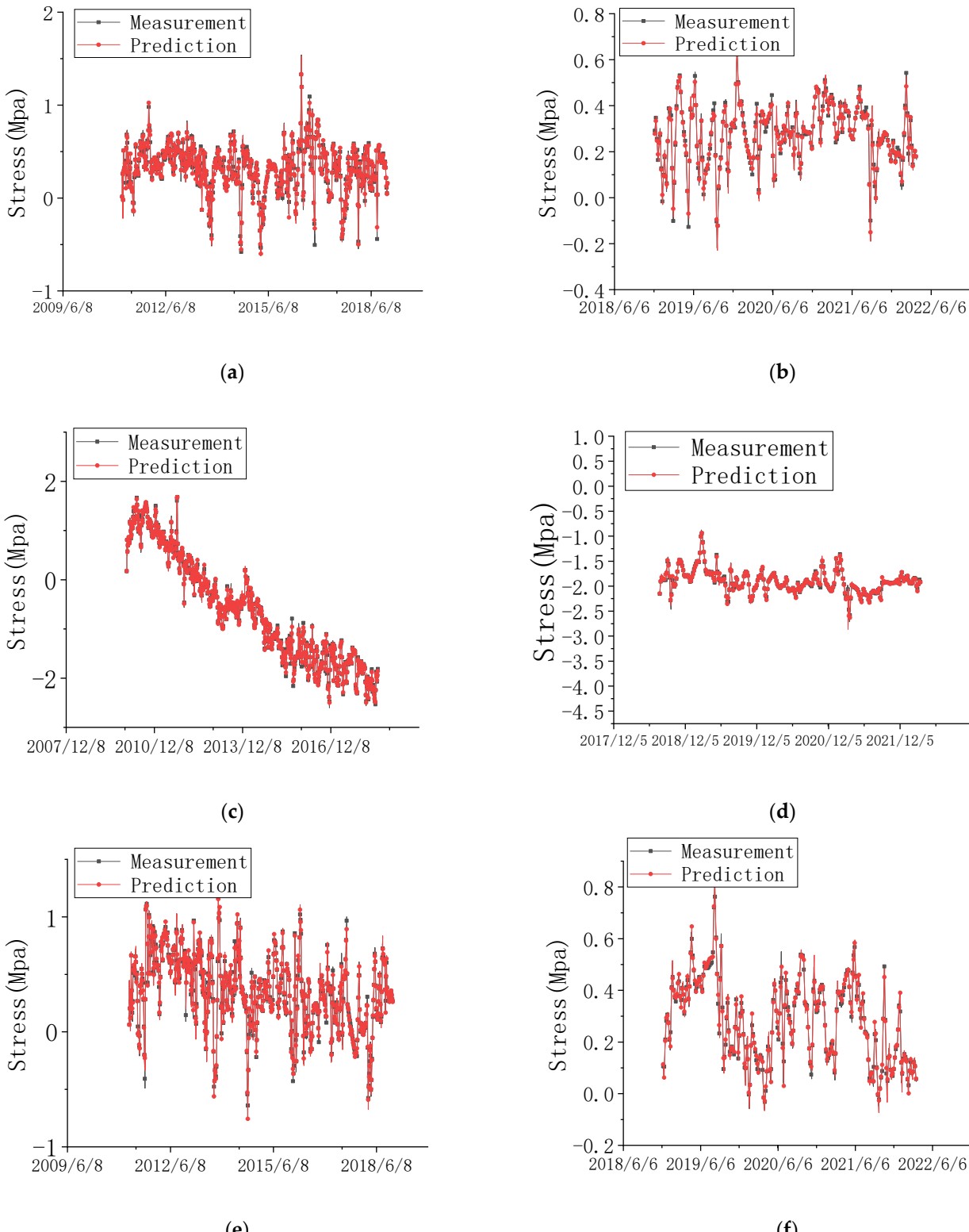

**Figure 9.** Study point EPR model results. (**a**) S67-1 stress training set model results; (**b**) S67-1 stress prediction set model results; (**c**) S616-4 stress training set model results; (**d**) S616-4 stress prediction set model results; (**e**) S622-5 stress training set model results; (**f**) S622-5 stress prediction set model results.

**Table 1.** EPR model evaluation index (unit: Mpa).

| Points | MAE | | RMSE | |
|---|---|---|---|---|
| | **Training Set** | **Testing Set** | **Training Set** | **Testing Set** |
| S67-1 | 0.01568 | 0.01229 | 0.02619 | 0.019378 |
| S616-4 | 0.01666 | 0.02916 | 0.02665 | 0.216171 |
| S622-5 | 0.01678 | 0.01070 | 0.03055 | 0.016792 |

*4.3. Multi-Model Comparison Results*

In this section, we have utilized standard GPR, LSTM, and SVR models as benchmark comparisons against our proposed EPR model. The evaluation of these models is based on the MAE and RMSE metrics, providing a means to assess the effectiveness of the EPR model presented in this study.

Figure 10a–f present a comparative analysis involving standard GPR, EPR, and measured values. In these visualizations, the measured values are represented in black, the EPR model's predictions in red, and the GPR model's predictions in blue. Notably, these figures highlight that the utilization of decomposed stress data significantly improves prediction accuracy within the model. Furthermore, the integration of the PSO model appears to yield favorable enhancements in parameter optimization outcomes.

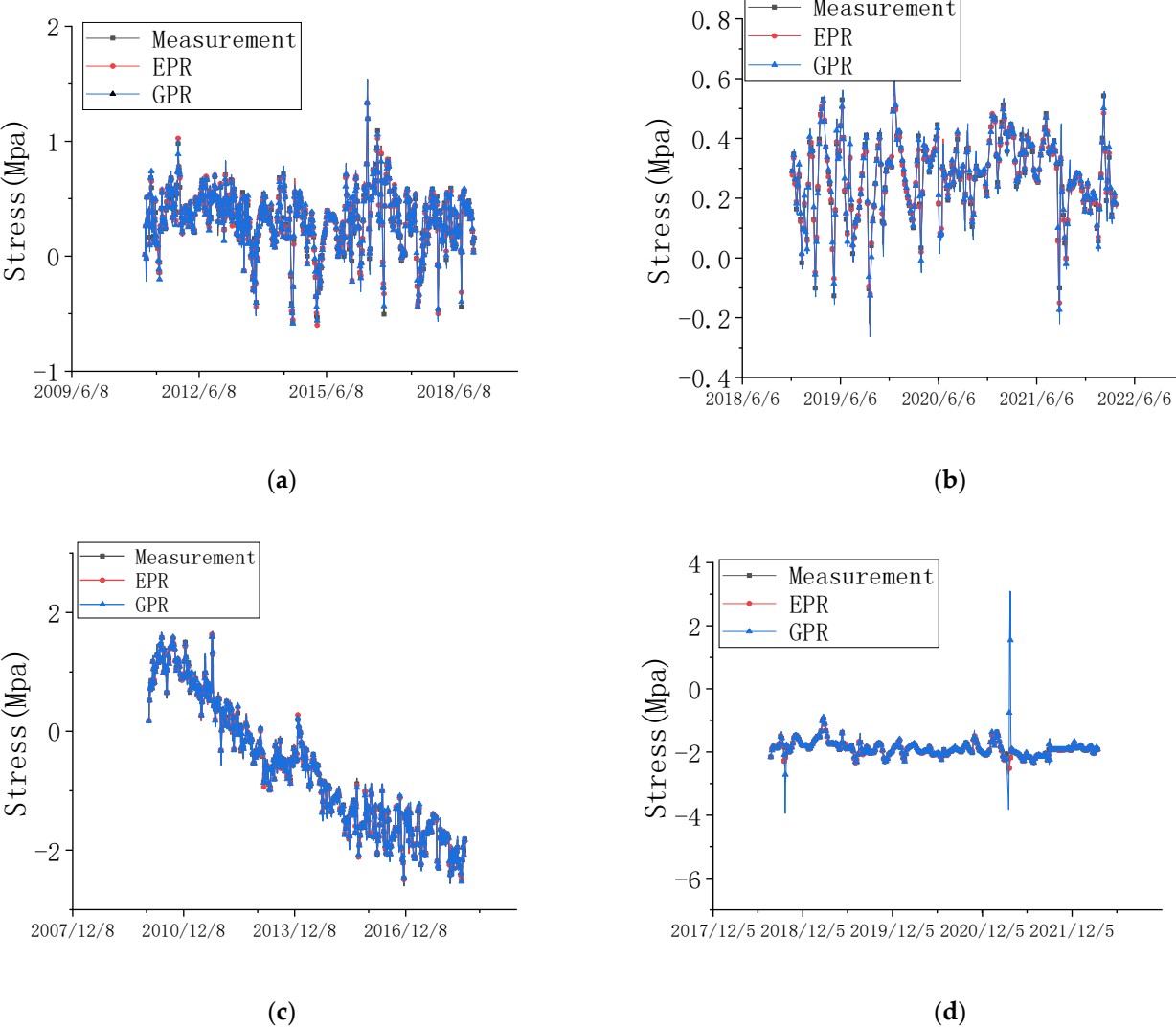

(**a**)

(**b**)

(**c**)

(**d**)

**Figure 10.** *Cont.*

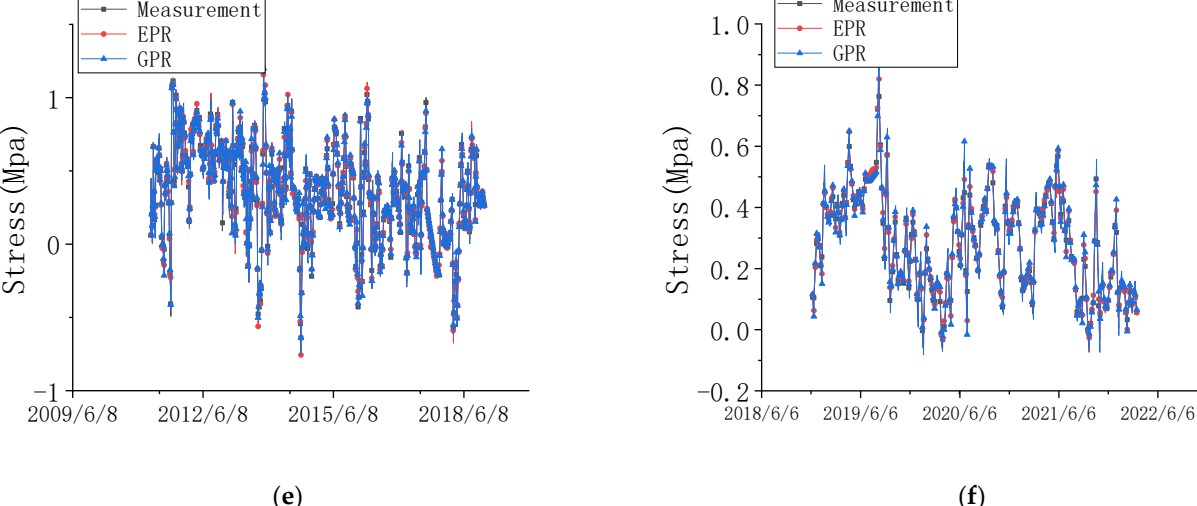

(**e**)                                                                     (**f**)

**Figure 10.** Comparison chart of EPR model and GPR model results. (**a**) S67-1 stress training set model results; (**b**) S67-1 stress prediction set model results; (**c**) S616-4 stress training set model results; (**d**) S616-4 stress prediction set model results; (**e**) S622-5 stress training set model results; (**f**) S622-5 stress prediction set model results.

To further validate the effectiveness of our model, this section conducts a comparison between the results obtained from the widely used LSTM and SVR models, both commonly employed in time series forecasting, and the results from our proposed EPR model. Figure 11 illustrates the results of the calculations. The illustration showcases the comparative results of the LSTM, SVR, and EPR models. The color scheme designates black for the measured values, red for the EPR's predictions, blue for the SVR's predictions, and green for the LSTM's predictions. A clear observation from the illustration is that the predictive capability of the EPR model outperforms both the LSTM and SVR models.

Table 2 presents the evaluation indices of different models at the three research points. A noteworthy observation from the table is that the proposed EPR model exhibits the smallest index values. Furthermore, the non-optimized GPR model performs quite well when compared to traditional time series models. This observation underscores the suitability of GPR for processing high-frequency data such as stress monitoring data. The incorporation of EMD and PSO further enhances the competence of GPR in handling high-frequency and high-noise data.

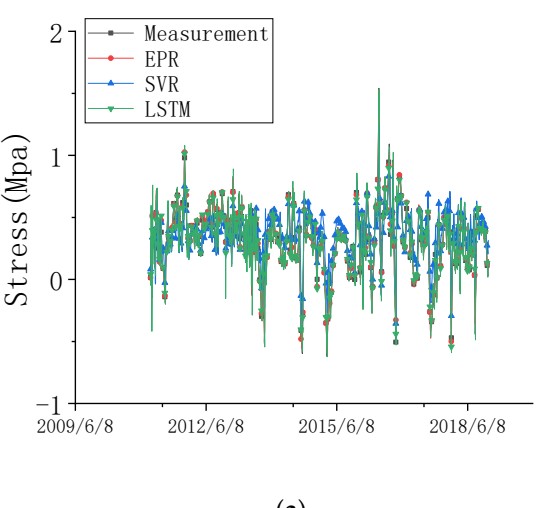

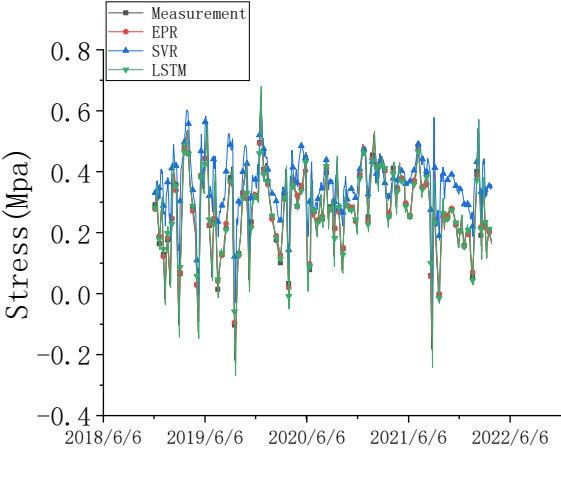

(**a**)                                                                     (**b**)

**Figure 11.** *Cont.*

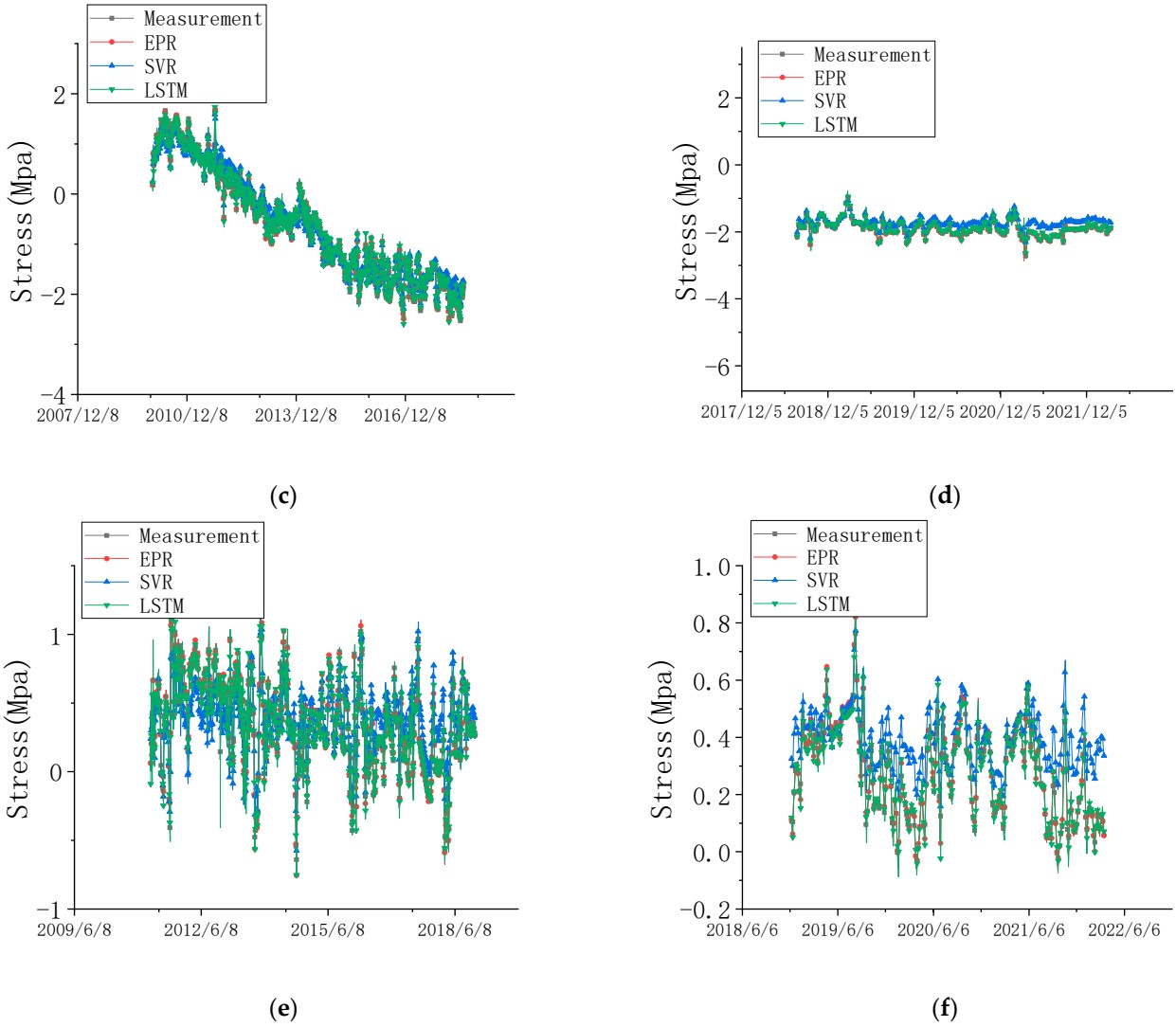

**Figure 11.** Comparison chart of LSTM model, SVR model, and EPR model. (**a**) S67-1 stress training set model results; (**b**) S67-1 stress prediction set model results; (**c**) S616-4 stress training set model results; (**d**) S616-4 stress prediction set model results; (**e**) S622-5 stress training set model results; (**f**) S622-5 stress prediction set model results.

**Table 2.** Evaluation index values of different models.

| Points | Models | MAE | | RMSE | |
|---|---|---|---|---|---|
| | | **Training Set** | **Testing Set** | **Training Set** | **Testing Set** |
| S67-1 | EPR | 0.015681 | 0.012295 | 0.02619 | 0.019378 |
| | GPR | 0.017645 | 0.013530 | 0.032534 | 0.023581 |
| | SVR | 0.110314 | 0.093430 | 0.133079 | 0.110845 |
| | LSTM | 0.027909 | 0.015469 | 0.047819 | 0.023698 |
| S616-4 | EPR | 0.016662 | 0.029167 | 0.026659 | 0.216171 |
| | GPR | 0.017858 | 0.046568 | 0.033340 | 0.299895 |
| | SVR | 0.120232 | 0.193641 | 0.147590 | 0.215437 |
| | LSTM | 0.03212 | 0.018291 | 0.052172 | 0.035915 |
| S622-5 | EPR | 0.016785 | 0.010702 | 0.030551 | 0.016792 |
| | GPR | 0.018956 | 0.012213 | 0.038120 | 0.022059 |
| | SVR | 0.137680 | 0.118263 | 0.165617 | 0.143403 |
| | LSTM | 0.032729 | 0.015945 | 0.058211 | 0.023396 |

However, the performance of the SVR model is lackluster. This can be primarily attributed to the fact that SVR is data-driven, and its fitting calculations for stress values in super-high arch dams rely on linear principles that inadequately account for the inherent nonlinearity in the stress patterns of super-high arch dams. Fixed parameters and static calculations further misalign with the operational principles of super-high arch dams, resulting in significant errors.

## 5. Conclusions

In conclusion, this paper has introduced and investigated the EPR model as an effective method for stress prediction in super-high arch dams. The key findings and contributions of this study can be summarized as follows:

Advantages of the EPR model: the proposed EPR model combines the strengths of GPR and EMD models. It decomposes high arch dam stress data into components with varying frequencies, effectively handling nonlinearity, randomness, and noise in the data. This enhances the GPR model's ability to analyze stress data from high arch dams. On the same test set, the EPR model's accuracy is maximally improved by 87% compared to the comparison model. This further indicates that the EPR model proposed in this paper has a better capability in dealing with high arch dam strain data and can be better adapted to engineering problems.

Role of PSO model: the inclusion of the PSO model plays a critical role in parameter optimization when processing stress data with the GPR model. Given that stress data may deviate from a strict Gaussian distribution and high-frequency EMD components lack a precise functional distribution, traditional maximum likelihood estimation for parameter estimation falls short. The PSO model helps identify optimal model parameters, thereby improving prediction accuracy.

Dynamic learning aspect: The EPR model distinguishes itself from static baseline models by introducing dynamic learning. It can adapt to changing parameters and the inherent nonlinearity in super-high arch dam stress monitoring data. This incorporation of dynamic features strengthens the model's predictive capabilities. The introduction of dynamic modeling remains crucial for the development of the field of hydraulic monitoring since the real working environment of extra-high arch dams as well as their own parameters are changing all the time.

Robustness and applicability: the EPR model's success in predicting stress across different sections of the dam demonstrates its robustness. This suggests that the proposed model can be extended to address stress prediction challenges in other ultra-high arch dams.

Looking forward, future research will focus on integrating spatial dimension information learning into ultra-high arch dam monitoring data. This will tailor the model to the unique characteristics of extra-high arch dams, further enhancing prediction accuracy. The EPR model represents an innovative approach to developing stress prediction models for high arch dams.

**Author Contributions:** Conceptualization, Y.W., C.H. and X.Z.; methodology, Y.W. and C.H.; software, Y.W.; validation, Y.W., C.H. and X.Z.; formal analysis, Y.W.; investigation, C.H. and Y.W.; resources, C.H., H.Z., D.T. and Y.Z.; data curation, Y.W. and C.H.; writing—original draft preparation, Y.W.; writing—review and editing, Y.W. and Y.H.; visualization, Y.H.; supervision, Y.W.; project administration, Y.H.; funding acquisition, C.H., H.Z., D.T. and Y.Z. All authors have read and agreed to the published version of the manuscript.

**Funding:** The research was funded by the National Natural Science Foundation of China (No. 51839007) and the Xiluodu technology development contract (No. Z412202008).

**Data Availability Statement:** Data are contained within the article.

**Acknowledgments:** The authors are grateful for the financial support of from the Funders.

**Conflicts of Interest:** Authors Chunyao Hou, Hongyi Zhang, Dawen Tan, Yi Zhou were employed by the company Three Gorges Jinsha River Chuanyun Hydropower Development Co., Ltd. The

remaining authors declare that the research was conducted in the absence of any commercial or financial relationships that could be construed as a potential conflict of interest.

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
