# Peer review of "Stress Prediction Model of Super-High Arch Dams Based on EMD-PSO-GPR Model"

_water, doi:10.3390/w15234087_

Round 1

Reviewer 1 Report (New Reviewer)

Comments and Suggestions for Authors

In this paper the prediction of the internal stresses for super-high arch dams have been estimated. For this, several numerical analyses have been carried out. The paper and theme appear very interesting theferore it is valid for pubblication.

Only some adjustments are necessary:

- Please, revise the Abstract since many acronyms are present. Define EPR, which is a key model in the paper.

- Introduction: revise the text. Why are there parts in red?

- Please, revise all equations and parameters.

- Section 2: Is it a new model? Maybe you can name this section as "Proposed model". Section 3 can be named "Data collection and pre-analyses". Revise the whole structure of the paper.

- Results (Figs. 5-7) in Section 3.2 appear very interesting. Please, provide more details and explications. 

- Provide more details in Conclusions.

Possible references:

a) https://doi.org/10.1016/j.soildyn.2016.09.039

b) https://doi.org/10.1007/s40996-018-0223-z

c) https://doi.org/10.1016/j.applthermaleng.2010.05.027

d) https://doi.org/10.1016/j.soildyn.2016.09.011

Author Response

In this paper the prediction of the internal stresses for super-high arch dams have been estimated. For this, several numerical analyses have been carried out. The paper and theme appear very interesting theferore it is valid for pubblication.

Only some adjustments are necessary:

- Please, revise the Abstract since many acronyms are present. Define EPR, which is a key model in the paper.

REPLY: We have revised the issue of the abbreviation of EPR in the abstract.

- Introduction: revise the text. Why are there parts in red??

REPLY: Changes have been made in red.

- Please, revise all equations and parameters.。

REPLY: We have proofread all the formulas in this paper.

- Section 2: Is it a new model? Maybe you can name this section as "Proposed model". Section 3 can be named "Data collection and pre-analyses". Revise the whole structure of the paper.

REPLY: We have revised the section headings of the article based on your suggestions.

- Results (Figs. 5-7) in Section 3.2 appear very interesting. Please, provide more details and explications. 

REPLY: We have added a description of Figs. 5-7 to make the article more specific in its explanation.

- Provide more details in Conclusions.

REPLY: The conclusions were revised.

Possible references:

  1. a) https://doi.org/10.1016/j.soildyn.2016.09.039
  2. b) https://doi.org/10.1007/s40996-018-0223-z
    c) https://doi.org/10.1016/j.applthermaleng.2010.05.027
  3. d) https://doi.org/10.1016/j.soildyn.2016.09.011

REPLY: We have supplemented the article by adding some references.

Reviewer 2 Report (New Reviewer)

Comments and Suggestions for Authors

The present work is concerned with a hybrid model model for predicting the lifespan of super-high arch dams, based on Empirical Mode Decomposition, Gaussian Process Regression, and Particle Swarm Optimization. Three strategically selected measurement points within the dam are chosen to evaluate stress conditions. The predicted results from the proposed method are compared with some well-known others, using actual stress data measured at research points within a super-high arch dam in southwest China, and the results reveal that the given model here attains a better maximum mean absolute errors. Incorporation of deep learning models into the realm of stress prediction for super-high arch dams constitutes the innovation of the method, which can be beneficial in dam safety monitoring.

The work seems to be publishable in MDPI: Water if a careful attention is given to the following coments;

A)- In Abstract, all abbreviations should be expanded. What is EPR?

B)- Language is fine overall, take care of typos and some little grammar.

C)- The literature survey is up to date and well-designed. Some more recent attempts from the current journal can be added.

D)- Provide reference(s) for the used equations, if they are not proposed for the first time in the present work. 

E) – Some other series methods can also be used for the current purposes, as referred to “Nonlinear Problems via a Convergence Accelerated Decomposition Method of Adomian (DOI: 10.32604/cmes.2021.012595)”.

F)- Calculation of true stress ig given through the series in (10). How to assure the convergence of series if n is large?

G)- Provide a Table for the parameters of Case Study.

H)- Correct Eq.(24).

K)- I think IMF6 and IMF7 are less physical. Please discuss.

M) – Looking into future research, it will focus on integrating spatial dimension information learning into ultra-high arch dam monitoring data. Are you going to use a continuous model in this case. If so, please state it explicitly.

Comments on the Quality of English Language

Good.

Author Response

The present work is concerned with a hybrid model model for predicting the lifespan of super-high arch dams, based on Empirical Mode Decomposition, Gaussian Process Regression, and Particle Swarm Optimization. Three strategically selected measurement points within the dam are chosen to evaluate stress conditions. The predicted results from the proposed method are compared with some well-known others, using actual stress data measured at research points within a super-high arch dam in southwest China, and the results reveal that the given model here attains a better maximum mean absolute errors. Incorporation of deep learning models into the realm of stress prediction for super-high arch dams constitutes the innovation of the method, which can be beneficial in dam safety monitoring.

The work seems to be publishable in MDPI: Water if a careful attention is given to the following coments;

A)- In Abstract, all abbreviations should be expanded. What is EPR?

 REPLY: We have revised the issue of the abbreviation of EPR in the abstract.

B)- Language is fine overall, take care of typos and some little grammar.

 REPLY: We have revised the full text.

C)- The literature survey is up to date and well-designed. Some more recent attempts from the current journal can be added.

REPLY: We have supplemented the text with some additional references.

D)- Provide reference(s) for the used equations, if they are not proposed for the first time in the present work. 

REPLY: We calibrated the full text formulas and added some literature.

  1. E) – Some other series methods can also be used for the current purposes, as referred to “Nonlinear Problems via a Convergence Accelerated Decomposition Method of Adomian (DOI: 10.32604/cmes.2021.012595)”.

REPLY: ADM method is for data with specific mathematical equation form. Since the processing object in this paper is the actual monitoring strain data in the real environment, which does not have a specific mathematical equation form, it is not applicable with the ADM method. The EMD method used in this paper is an efficient way to process the signal, which can decompose the actual data according to the specific data rules. Therefore EMD is more suitable for the data in this paper.

F)- Calculation of true stress ig given through the series in (10). How to assure the convergence of series if n is large?

REPLY:  is the inverse of the equivalent modulus of elasticity, E is the modulus of elasticity of concrete,  is the degree of creep, and  is the age of onset loading. The physical phenomenon of creep occurs early in the deformation of concrete. When  is large, E is not enhanced with time,  tends to 0, and the inverse of the equivalent modulus of elasticity tends to a constant. where  are all the strain increments converted from the current monitored strain minus the strain occurring before the corresponding time period of , i.e., the  time period, i.e., the strain increment of the current time period.

G)- Provide a Table for the parameters of Case Study.。

REPLY: Our EPR model only needs to set the PSO model optimization objective, i.e., set the upper and lower bounds of the objective, and the model parameters are all obtained according to the model training, and there is no need to set the parameters in advance, which is one of the advantages of this model.

H)- Correct Eq.(24).

 REPLY: We recalibrated the full formula

K)- I think IMF6 and IMF7 are less physical. Please discuss.

REPLY: The EMD method is to transform the high-frequency data into signals of different frequencies and then do the processing. If the strain data are not processed, the model may ignore some of the information, such as IMF6 and IMF7. In this paper, the EMD method is used in order not to reduce the model accuracy by having some of the information in the data rejected by the computational model.

  1. M) – Looking into future research, it will focus on integrating spatial dimension information learning into ultra-high arch dam monitoring data. Are you going to use a continuous model in this case. If so, please state it explicitly.

REPLY: The follow-up work will be centered on continuous field output, but the specific research methodology has not yet been determined, so it is not developed in the text to avoid misleading the reader about the work。

This manuscript is a resubmission of an earlier submission. The following is a list of the peer review reports and author responses from that submission.

Round 1

Reviewer 1 Report

Comments and Suggestions for Authors

Comments on the Quality of English Language

Author Response

We have touched up the language throughout the text.

Reviewer 2 Report

Comments and Suggestions for Authors

The document is too long for the content: It can be much shorter.

-Some authors have a star behind their name, which is not explained at all: where does this star refer to? If this is intended to refer to corresponding authorship (it is strange that there are 2): the work email MUST be used.

-I also have the impression that the last names are given first, the first names second. This is very uncommon in western literature. The authors are advised to reconsider.

-The authors have missed mentioning very important literature:

Zhou  et al: Study on regression analysis and simulation feedback-prediction methods of super high arch dam during construction and first impounding process, Earth and Space 2012: Engineering, Science, Construction, and Operations in Challenging Environments, 2012.

Li et al: Smart monitoring of a super high arch dam during the first reservoir-filling phase, Journal of Aerospace Engineering, 2017.

Lin et al: Horizontal cracking and crack repair analysis of a super high arch dam based on fracture toughness, Engineering Failure Analysis, 2019.

-Figures 5 and 6 are too small: I can not read the text.

-The models are described at length in section 2, however, not much reference is made to this in the remainder of the paper. This must be improved (in connection to the results), otherwise section 2 can be considerably shortened or removed.

-Figures 8, 10, and 11: y too small. If this is the figure you want to display: simply delete it, it is useless.

Figure 3 (not the real figure 3, but the second figure 3 on page 17): same as for Figure 8. Also, this error in figure numbering should have been noticed during proofreading.

To conclude: the manuscript must be significantly improved if this is to be considered for publication. In the current state, it more reads as a project report, focusing on some technical details and leaving the science part wide open. The authors need to revise the paper in a major way.

Comments on the Quality of English Language

-

Author Response

1.Some authors have a star behind their name, which is not explained at all: where does this star refer to? If this is intended to refer to corresponding authorship (it is strange that there are 2): the work email MUST be used.

Reply:Dear Editor, A star after an author's name means that he or she is a corresponding author. This paper uses dual corresponding authors.

2.I also have the impression that the last names are given first, the first names second. This is very uncommon in western literature. The authors are advised to reconsider.

Reply:We have modified it.

3.The authors have missed mentioning very important literature:

Reply:We have added relevant literature at appropriate places in the paper.

4.Figures 5 and 6 are too small: I can not read the text.

Reply:We have modified the image format.

5.The models are described at length in section 2, however, not much reference is made to this in the remainder of the paper. This must be improved (in connection to the results), otherwise section 2 can be considerably shortened or removed.

Reply:Dear reviewer. Since the model presented in this paper requires almost no specialised setting of hyperparameters, it has not been specifically stated in the example.

6.Figures 8, 10, and 11: y too small. If this is the figure you want to display: simply delete it, it is useless.

Reply:We have modified the image format.

7.Figure 3 (not the real figure 3, but the second figure 3 on page 17): same as for Figure 8. Also, this error in figure numbering should have been noticed during proofreading.

Reply:We have modified the image format.

To conclude: the manuscript must be significantly improved if this is to be considered for publication. In the current state, it more reads as a project report, focusing on some technical details and leaving the science part wide open. The authors need to revise the paper in a major way.

Thank you very much for your valuable comments on this article, we have revised the entire text according to your suggestions.

Reviewer 3 Report

Comments and Suggestions for Authors

Dear authors! I am sending comments on your manuscript. Best regards.

Author Response

  1. Formula numbers are missing (4), (10).

We have modified it。

  1. Incorrect numbering of Figure 3 (instead of Figure 3, Figure 1 is printed), also Figure

7 (instead of Figure 7, Figure 2 is printed), also Figure 9 (instead of Figure 9, Figure 3 is  printed). 

We have proofread the full text.

  1. When assessing the strength of concrete in an arch dam, the values of the main stresses (maximum and minimum) are used, compared with the permissible compressive and tensile stresses. The article (Figure 7) presents the results of  measurements at the considered points of the values of normal stresses in accordance  with the directions of the coordinate axes. How the transition to the main voltages is  made.  

In analysing the damage, there are four classical damage criteria: maximum tensile stress, maximum tensile strain, maximum shear stress, strain energy density, etc., and we refer to the maximum tensile stress criterion; Fig. 7 illustrates the case where the monitoring results of the strain gage set are resolved into stresses, and we will compute the indicator of maximum principal stresses based on these results. We also add the corresponding content into the article.

  1. It is necessary to represent the directions of the coordinate axes in the Figure 3.

We have modified it。

  1. The observation period from 02.2011 to 05.2022 is presented. Which part of the period refers to the construction period, and which to the operational period?

The period 2011 - March 2014 is the construction period and the rest is the storage period.

  1. The results of field observations at three points in the volume of an ultra-high arch dam are presented, which is completely insufficient for such a responsible structure. It is advisable to add information about the actual number of points of measurements, if they  were carried out.  

There are several reasons for this paper to select these three measurement points as the research object: ① The theme of this paper is to propose a general stress prediction model, and the Xiluodu project is chosen as a practical case, so when selecting the measurement points, a limited number of measurement points are chosen for in-depth analysis according to the characteristics of the project; ② Xiluodu (Ref. The phenomenon of the valley contraction unique to this project, as well as the existing monitoring data, suggests that the state of the stresses at the base of the dam should be considered as a priority; ③ Xiluodu instrumentation in the vicinity of the base of the dam is extremely centralised, and the concentration of the instruments, such as the multi-point displacement meter and the reinforcement meter, is conducive to the analysis.

  1. What determines the choice of full-scale measurement points presented in the article? All points are located at the foot of the arch dam. In what part of the thickness of the dam? In the center or near the surface? From the upstream or downstream side? The  influence of factors on deformations and stresses depends on this. In arch dams, the  occurrence of maximum stresses is also possible in other zones. It depends on many  factors: the geometry of the dam, the properties of the base, etc.  

In this paper, the measurement points at the dam heel area near the upstream side were selected. Due to the property that concrete cannot be tensile, this paper mainly selects three representative measurement points of the monitoring data of this extra-high arch dam to illustrate the calculation ability of this model.

  1. It would be useful to present in the article the main parameters and features of the structure in addition to the height of the arch dam: the length along the crest, the width of the overlapped canyon, the thickness at the crest and base.   

We have made changes in the appropriate parts of the text,

  1. In Figures 9, 10, 11, it is not clear which stress are represented? Normal? What are the stress components?

The figure represents the maximum principal stresses for the model solution

Round 2

Reviewer 1 Report

Comments and Suggestions for Authors

Manuscript ID: water-2522973
Title: Stress prediction model of super-high arch dams based on EMD-PSO-GPR model.
OVERVIEW
The authors have improved the manuscript and in my opinion, it may be published as it is.  

Comments on the Quality of English Language

The English is fine. A final proofreading is recommended.

Author Response

Dear reviewer, Thank you very much for reviewing and guiding this paper. We have proofread the full paper.

Reviewer 2 Report

Comments and Suggestions for Authors

I feel that the authors have not sufficiently considered my previous remarks/review in their revision. Frankly, i doubt that the authors have even understood some of my questions... I do feel the authors have onlz changed some issues with the paper that were "easy wins", whereas to the other remarks rather vague or answers that are not relevant.... Therefore, I can only advise the editor to reject the paper as I see hardly any improvement. 

Comments on the Quality of English Language

-

Author Response

Dear reviewer:

Hello, thank you very much for your valuable comments on our paper. We have made another in-depth revision in response to your previous questions. The revisions are as follows:

1、 We have added the important literature you proposed to the appropriate position. In the process of writing the article, because this paper mainly studies the establishment of monitoring models in stress, so it does not consider the citation of some articles on deformation models. Moreover, the special characteristics of XLD arch dam were neglected in the writing process, and no relevant literature was cited to illustrate the special characteristics of the project. Therefore, after your valuable suggestions, we supplemented the references appropriately and added the literature you suggested to the appropriate position.

  1. We have adjusted and appropriately deleted the second chapter. We have adjusted the order of the second chapter and deleted the redundant parts.
  2. We have resized the text of the horizontal and vertical headings of the article's graphs and charts to make them easier to read.

Finally, we would like to thank you once again for taking the time to review this article.

Round 3

Reviewer 2 Report

Comments and Suggestions for Authors

The introduction still rather misses some important topics related to the current research: I think not enough introduction is given to topics that are relevant to the current topic at hand. Although maybe not really the topic of this investigation, nevertheless, such topics and/or alternative methods/approaches should be mentioned/discussed to give the reader (e.g. a starting PhD student) a proper and balanced introduction.

As this is already the 3rd review round, and you do not provide a revision that would allow me to change my opinion (which is still reject), i advise the editor to either make a judgement call or send the submitted paper out to an additional review by a third party.

Comments on the Quality of English Language

-